# Analysis of the Composite Risk Grade for Multi Extreme Climate Events in China in Recent 60 Years

Cunjie Zhang [1,*], Chan Xiao [1], Shuai Li [2], Yuyu Ren [1], Siqi Zhang [1], Xiuhua Cai [3] and Zhujie Sangbu [4]

1   CMA National Climate Centre, Beijing 100081, China; xiaochan@cma.gov.cn (C.X.);
    renyuyu@cma.gov.cn (Y.R.); zhangsq@cma.gov.cn (S.Z.)
2   China Three Gorges Corporation, Yichang 443133, China; li_shuai@ctg.com.cn
3   Chinese Academy of Meteorological Sciences, Beijing 100081, China; caixh@cma.gov.cn
4   Meteorological Bureau of Nanmulin County, Tibet Autonomous Region, Shigatse 857000, China;
    xzqxzjsb@163.com
*   Correspondence: zhangcj@cma.gov.cn; Tel.: +86-010-5899-3328

**Abstract:** Much attention has been given to the change rule of a single extreme event, and there are few reports on comprehensive characteristics of multiple extreme events in a certain region. Based on the analyzes of annual frequency of extreme high temperature, extreme low temperature, extreme drought, extreme precipitation, and extreme typhoon events in China from 1961 to 2020, a multi extreme events composite risk grade index (MXCI) was constructed and applied to the comprehensive characteristics analyzes of multiple extreme events in China. The results show that the high value areas of MXCI were mainly located in southeast China and southwest China. Over the past 60 years, the MXCI has shown a decreasing trend in western China and most of southeastern China, and an increasing trend in the middle zone from southwest China to northeast China. Through comparative analysis, MXCI can objectively reflect the comprehensive characteristics of multiple extreme climate events in a region, which is helpful to understand regional extreme climate characteristics and effectively cope with extreme climate risks.

**Keywords:** extreme climate events; multi extreme event composite grade index (MXCI); climate change; China





## 1. Introduction

In August 2021, the Intergovernmental Panel on Climate Change (IPCC) released the report "Climate Change 2021: The Physical Science Basis" [1], showing that the observed increase in average temperature in Asia has gone beyond natural variability, accompanied by an increase in extreme warm events and a decrease in extreme cold events, and this trend will continue in the future. The combustion of fossil fuels has led to climate change, giving rise to a cascade of extreme weather events. As a result, the production system is now grappling with profound and far-reaching consequences. Therefore, to proactively cope with frequent extreme weather and climate events and their resultant adverse socioeconomic impacts, the IPCC released a special report 'Managing the Risks of Extreme Events and Disasters to Advance Climate Change Adaptation (SREX)' in 2012 [2].

In recent decades, extreme weather and climate events such as droughts and floods also occurred frequently in China [3–7], resulting in increasingly serious socioeconomic losses. In the summer of 2020, severe rainstorm and flood events occurred in the Yangtze River Basin [8], and in July 2021, extreme heavy precipitation events occurred in many places of Henan Province [9]. There phenomena indicate that precipitation extremes and the impacts are becoming more severe. In the summer of 2022, a severe high-temperature drought event occurred in the Yangtze River Basin [10], causing serious impacts on social and economic development. Some Studies [11–13] have shown that there has been an increasing trend of northward-moving typhoons affecting China in recent years.

To fully absorb the latest scientific advances at home and abroad and draw lessons from the advanced concepts and experiences of the international community in extreme event research and disaster risk management, Qin et al. [3] organized the compilation of the report 'China National Assessment Report on Risk Management and Adaptation of Climate Extremes and Disasters (CSREX)', which sufficiently reflects the characteristics of China's disaster prevention and mitigation system. The report pointed out that in the context of global climate change, the risk of meteorological disasters in China has been further exacerbated, and the situation of disaster prevention and mitigation is exceptionally serious. Furthermore, Qin and Zhai organized the compilation of the report 'Evolution of China's Climate and Ecological Environment (2021)' [4], which assessed the variation patterns of extreme temperature, extreme precipitation, droughts, typhoons, and other climate events in China in recent decades. Using CMIP5 data, Wu et al. [14] analyzed the risk of future climate change in China on the basis of considering the extreme events of drought, heat wave, and flood. Many research reports focus on the change rule of a single extreme event [15,16], and there are few reports on comprehensive characteristics of multiple extreme events in a certain region, which is very important for understanding the characteristics of regional extreme climate and coping with climate risks.

To characterize the overall variations in extreme events in a region, many studies have proposed the composite index for extreme events. In response to the weather and climate characteristics of the United States, Karl et al. [17] proposed the Climate Extremes Index (CEI), a new index consisting of a combination of traditional characterization indexes for extreme events, to study the variations in extreme events in the United States. Ren et al. [18] selected seven extreme climate indexes with high economic and social impacts on Chinese mainland, i.e., the national-average, high-temperature days and low-temperature days, heavy precipitation days, dust storm days, strong wind days, drought-area percentage, and landed tropical cyclone frequency, which are then composited into two extreme event indexes (IECI-I and IECI-II). Furthermore, the overall evolution trend of extreme events over Chinese mainland from 1956 to 2008 was analyzed. The IECI-I and IECI-II are the equally and unequally weighted sums of the standardized extreme climate indexes. Specifically, for IECI-II, varied relative importance and weights are assigned to the single extreme climate index based on the economic losses and social impacts induced by that extreme event. Based on the historical meteorological disaster data collected by meteorological departments, Wang et al. [19] determined the weight coefficients of five extreme events (flood, drought, high temperature, low temperature, and typhoon) in each month to construct a Chinese climate risk index (CRI), this index has been applied to the annual report 'Blue Book on Climate Change in China' [6]. To address the issue of quantitative evaluation of natural disaster losses, Zhang et al. [20] proposed a calculation method for the composite disaster index ($I_R$) based on the geometric-mean model, which enables the quantitative calculation of multivariate disaster data for a given area.

To construct a composite extreme event index, the following issues need to be considered: First, if there are multiple extreme weather events in a region, which extreme events should be selected to construct the composite index? Second, what is the relationship between the composite extreme event index and individual extreme events? Is it an additive relationship, a multiplicative relationship, or some other relationship? Third, how to determine the proportion of individual extreme events in the composite extreme event index, that is, the weight factor?

In terms of selecting which kinds of extreme events to construct the comprehensive extreme index, the common method is to select the extreme events with strong extreme, high frequency, and great impact. The results of selection in different regions may be different. In this paper, five events of extreme high temperature, extreme low temperature, extreme drought, extreme precipitation, and extreme typhoon are selected to construct a comprehensive extreme event index for China.

The problem of how to determine the relationship between the composite index and a single extreme event is very complicated, and the common method is to use the

accumulation method. However, if two or more extreme climate events occur at the same time, there will be a positive feedback effect, such as extreme high temperature and extreme drought occur at the same time, the extreme and impact degree will increase, and the accumulation method may not be suitable for constructing the composite index. In this paper, composite events such as extreme heat and drought occur at the same time are not considered, and the composite index is still constructed using the accumulation method.

There have been studies on how to determine the weight factors when constructing the composite index. Ren's IECI [18], Wang's CRI [19], and Zhang's IR [20] all use the historical losses data to determine the weight factors of single extreme events. While this approach can reflect to some extent the severity of individual extreme events, it relies heavily on the quality and quantity of losses data. As we know, losses data is greatly affected by natural and man-made environmental factors at that time, and sometimes may not be accurate and objective. If it was used to determine the weight factor, the objectivity and extremity of the composite index will be affected.

This study analyzes the annual frequency of extreme high temperature, extreme low temperature, extreme drought, extreme precipitation, and extreme typhoon events in China from 1961 to 2020. On this basis, a composite grade index for extreme climate events is constructed by ranking and classifying the frequency in the whole nation with the percentile method. When determining the weights, this method determines the levels only based on the rank of the annual frequency of extreme events in China, which avoids the problem caused by the incomplete historical losses data. The higher the composite index grades, the higher the frequency of extreme events in the region, and the higher the climate risks.

Zhang's research [21] is the preliminary foundation work of this study, which introduces the construction process of the composite grads index of multiple extreme climate events and applies it to the Yangtze River basin. This study has improved the composite extreme climate risk index and added extreme typhoon events, extending the areas of the study to the whole region of China.

## 2. Methods and Data

### 2.1. The Discrimination Method for Extreme Events

In order to quantitatively distinguish extreme events from a mass of weather/climate data, a threshold needs to be determined. When a weather/climate record or variable exceeds the threshold, an extreme event is judged to have occurred. According to different threshold determination methods, the thresholds can be classified into "absolute" and "relative" categories.

The discrimination thresholds for extreme events calculated based on statistical probability analysis are relative thresholds, whose magnitude depends on the specific region and time period [22–24]. Relative thresholds tend to be more universal and comparable, and can more accurately reflect the extreme characteristics of climate in different regions and time periods. For instance, a heavy precipitation event with the same probability may have a relative threshold of 20 mm d$^{-1}$ in arid regions, but it may reach 80 mm d$^{-1}$ in humid regions. In addition, the use of relative thresholds in the studies of climate model simulations can effectively eliminate systematic bias in the simulation results. Therefore, relative thresholds are more commonly used as the criteria for discriminating extreme events in international academia.

The Joint Expert Team on Climate Change Detection and Index (ETCCDI) of the World Meteorological Organization Commission for Climatology has defined 27 typical climate indexes [25–28], namely 16 temperature indexes and 11 precipitation indexes. These indexes contain both absolute and relative threshold indexes. In order to characterize the different variations of extreme events, different indexes can be chosen flexibly. In recent years, these indexes have been widely used in the studies of extreme events such as extreme temperature and extreme precipitation [29–33].

Past studies have shown that weather and climate events such as heat waves, cold damage, droughts, heavy rainfall, and typhoons occur frequently in China and have a

significant impact on local social economy. To this study, these five events are selected for extremity analysis in this study. The relative threshold indexes determined by the percentile method are used as the discrimination indexes for extreme high temperature, extreme low temperature, extreme drought, extreme precipitation, and extreme typhoon (Table 1). Specifically, the Standardized Precipitation Index (SPI) of precipitation in the past 90 days is used for extreme drought in this study [34]. The precipitation of 3 consecutive days is taken as the study object of extreme precipitation, and an absolute threshold of no less than 30 mm is considered. Extreme high temperature and extreme low temperature are studied based on daily maximum temperature and daily minimum temperature, respectively. The precipitation and wind speed caused by typhoons (tropical cyclones) are identified and separated from those caused by other weather systems and are further analyzed separately.

**Table 1.** Definition of extreme climate event indices.

| Name | Abbreviation | Definition | Unit |
|---|---|---|---|
| Extreme high-temperature event | Eht | Daily maximum temperature exceeding 90th percentile | day |
| Extreme low-temperature event | Elt | Daily minimum temperature below 10th percentile | day |
| Extreme drought event | Edr | The SPI value for the last 90 days below −1.5 | day |
| Extreme precipitation event | Epr | Precipitation exceeding 90th percentile and not less than 30 mm for three consecutive days | day |
| Extreme typhoon precipitation | Etp | Daily precipitation of typhoon exceeding 90th percentile | mm |
| Extreme typhoon wind speed | Etw | Daily maximum wind speed of typhoon exceeding 90th percentile | m/s |

*2.2. Determination of the Relative Threshold*

In order to identify the extreme event, a corresponding threshold needs to be given. In this study, we refer to Folland et al. [35] and Bonsal et al. [36] for the threshold calculation method and use the Jenkinson empirical formula (Equation (1)) to calculate the threshold; the formula does not require the data to conform to a specific distribution. Folland compared different calculation methods to demonstrate the reliability of the Jenkinson formula in moderate extreme cases. The Jenkinson empirical formula is as follows:

$$CF = (M - 0.31)/(N + 0.38) \tag{1}$$

where *CF* is the cumulative frequency. In this study, extreme high temperature and extreme precipitation adopt the 90th percentile threshold (*CF* is 0.9), and extreme low temperature adopts the 10th percentile threshold (*CF* is 0.1). The variable *N* is the total number of samples participating in the calculation, and *M* is the sequence number corresponding to the threshold, according to which the specific threshold can be obtained. The constants −0.31 and 0.38 are empirical parameters.

*2.3. Discrimination of Extreme High Temperature (Low Temperature) Events*

The relative threshold method is used in this study to identify extreme high temperature (low temperature) events. Based on the percentile method, the thresholds at which the cumulative frequency of daily maximum (minimum) temperature is 90% (10%) are calculated and determined. Then, the days exceeding (below) the 90% (10%) threshold are regarded as extreme high-temperature (low-temperature) events. In order to increase the number of samples, the sequences of 5 days before and after are used to form the sequence of the day, and the new sequence is sorted to determine the thresholds.

The formula for discriminating extreme high-temperature events is as follows:

$$E_{ht} = \begin{cases} 1, & CF_{ht} \geq 90\% \\ 0, & CF_{ht} < 90\% \end{cases}. \tag{2}$$

The formula for discriminating extreme low-temperature events is as follows:

$$E_{lt} = \begin{cases} 0, & CF_{lt} > 10\% \\ 1, & CF_{lt} \leq 10\% \end{cases}. \tag{3}$$

In Equations (2) and (3), $CF_{ht}$ ($CF_{lt}$) is the cumulative frequency calculated from the samples. $E_{ht}$ ($E_{lt}$) is the extreme high-temperature (low-temperature) event, and 1 means it reaches extreme high-temperature (low-temperature) event and 0 means does not reach.

*2.4. Discrimination of Extreme Drought Events*

Here, the SPI of the precipitation in the past 90 days ($SPI_{90}$) is selected as the drought index, and the days above severe drought ($SPI_{90} \leq -1.5$) are taken as extreme drought events for the following statistical analysis. The formula for discriminating extreme drought events is as follows:

$$E_{dr} = \begin{cases} 1, & SPI_{90} \leq -1.5 \\ 0, & SPI_{90} > -1.5 \end{cases} \tag{4}$$

where $E_{dr}$ is the extreme drought event, and 1 means it reaches and 0 means does not reach.

*2.5. Discrimination of Extreme Precipitation Events*

Since daily precipitation in arid and semi-arid areas does not obey normal distribution, extreme precipitation events determined using the percentile method are sometimes a lot, but the rainfall of such events is often not heavy enough to cause rainstorms and flood disasters. Xie et al. [37] found that it is realistic to adopt the daily precipitation of 25 mm rather than 50 mm as the daily precipitation threshold for the occurrence of flood process in the Yunnan region. According to Wu et al. [38], the losses caused by flood events are closely related to the amount of precipitation in 3 consecutive days, and 30 mm is taken as the starting precipitation for the occurrence of flood events. Consequently, the precipitation of 3 consecutive days (R3) is adopted as the index for judging extreme precipitation, and the $R3 \geq 30$ mm is used as the constraint.

The relative threshold method is used in this study to determine extreme precipitation events. Based on the percentile method, the threshold at which the cumulative frequency of precipitation is 90% is calculated, and the events exceeding the 90% threshold are regarded as extreme precipitation events. In order to make the threshold stable, the number of samples needs to be increased. Here, the sequences of 5 days before and after are used to form the sequence of the day, and the new sequence is sorted to determine the threshold. The formula for discriminating extreme precipitation events is as follows:

$$E_{pr} = \begin{cases} 1, & CF_{R3} \geq 90\% \ and \ R3 \geq 30 \text{ mm} \\ 0, & CF_{R3} < 90\% \ or \ R3 < 30 \text{ mm} \end{cases} \tag{5}$$

where $R3$ is the precipitation of 3 consecutive days (mm), $CF_{R3}$ is the cumulative frequency calculated from the sample consisting of historical data (5 days before and after). The variable $E_{pr}$ is the extreme precipitation event, and 1 means it reaches and 0 means does not reach.

*2.6. Discrimination of Extreme Typhoon Events*

To analyze extreme typhoon events, it is necessary to discern and separate the wind and precipitation caused by typhoons. Ren et al. [39,40] developed an objective method for separating typhoon precipitation, namely the Objective Synoptic Analysis Technique (OSAT), which has a good performance in separating typhoon precipitation from the

precipitation caused by other weather systems. Yin et al. [41] developed a method for separating typhoon wind speed from the wind caused by other weather systems, and the typhoon wind speed is mainly determined based on the typhoon track and intensity. In this study, precipitation and wind speed caused by typhoons (including tropical cyclones) affecting Chinese mainland from 1961 to 2020 are separated by using the above two methods, and obtain the series of daily typhoon precipitation and daily maximum typhoon wind speed.

To make the selected thresholds stable, the same method as mentioned above is taken to increase the sample size; i.e., the values of 5 days before and after are used to form a new sequence for that day. Then, the sequence is sorted to determine the extreme thresholds. The percentile method is also used to determine the threshold where the cumulative frequency of precipitation is 90%. The events exceeding the 90% threshold are defined as extreme typhoon precipitation events ($E_{tp}$) or extreme typhoon wind speed events ($E_{tw}$), and the frequency of extreme typhoon events is obtained finally. The specific formula for the discrimination is as follows:

$$E_{ty} = \begin{cases} 2, & CF_{tp} \geq 90\% \ and \ CF_{tw} \geq 90\% \\ 1, & CF_{tp} \geq 90\% \ or \ CF_{tw} \geq 90\% \\ 0, & CF_{tp} < 90\% \ and \ CF_{tw} < 90\% \end{cases} \tag{6}$$

where $CF_{tp}$ and $CF_{tw}$ are the cumulative frequency calculated from the historical samples of daily typhoon precipitation and daily maximum typhoon wind speed of 5 days before and after, respectively, and $E_{ty}$ is the extreme typhoon event. $E_{ty}$ adopts 2, 1, and 0 for the cases when both daily typhoon precipitation and daily maximum typhoon wind speed exceed the threshold, only one reaches the threshold, and neither reaches the threshold, respectively.

*2.7. Determination of the Risk Level of a Single Extreme Event*

The extremity of a single extreme event needs to be judged in both temporal and spatial dimensions. The judgment in the temporal dimension refers to the comparison of the annual frequency of extreme events with their historical occurrence. The judgment in spatial dimension refers to the comparison of the frequency of extreme events at different regions. When examining the extremity levels of extreme events in China, it is necessary to compare the frequency of all stations on a national scale so that the risk level of that extreme event can be determined for each station.

The formulas for discriminating the risk level of a single extreme event are as follows:

$$R_x = \begin{cases} 4, & CF_{E_x} \geq 90\% \\ 3, & 70\% \leq CF_{E_x} < 90\% \\ 2, & 40\% \leq CF_{E_x} < 70\% \\ 1, & CF_{E_x} < 40\% \end{cases} \tag{7}$$

where $E_x$ is the annual frequency of one extreme event (such as $E_{ht}$ for extreme high-temperature events, $E_{lt}$ for extreme low-temperature events, $E_{dr}$ for extreme drought events, $E_{pr}$ for extreme precipitation events, and $E_{ty}$ for extreme typhoon events), $CF_{Ex}$ is the cumulative frequency of one extreme event and $R_x$ is the risk level determined by comparing the annual frequency of one extreme event nationwide, with 4, 3, 2, and 1 representing very high risk, high risk, medium risk, and low risk, respectively.

*2.8. Determination of the Composite Risk Level for Multi Extreme Climate Events*

To study the composite risk level of multi extreme climate events in a region, it is a requisite to determine the risk levels of individual extreme events, as well as the proportions of individual extreme events in the whole extreme events. Based on the analysis in Sections 2.2–2.7, the thresholds for the historical extremities of individual extreme events have been identified, and the risk levels have been assigned according to the percentile method. Hence, the composite risk grades of multi extreme events is

determined by simply adding up the risk grades of individual extreme events, which also solves the issue of comparability of multi extreme events in different areas within a region. The formulas for the composite risk grades of multi extreme events are as follows:

$$MXCI = R_{ht} + R_{lt} + R_{dr} + R_{pr} + R_{ty}, \tag{8}$$

$$\begin{cases} 4, \ CF_{MXCI} \geq 90\% \\ 3, \ 70\% \leq CF_{MXCI} < 90\% \\ 2, \ 40\% \leq CF_{MXCI} < 70\% \\ 1, \ CF_{MXCI} < 40\% \end{cases}. \tag{9}$$

In Equation (9), MXCI is the multi extreme events composite risk grade index and $R_{ht}$, $R_{lt}$, $R_{dr}$, $R_{pr}$, and $R_{ty}$ are the risk levels of extreme drought event, extreme precipitation event, extreme high-temperature event, extreme low-temperature event, and extreme typhoon event, respectively. In Equation (9), $CF_{\mathrm{MXCI}}$ is the cumulative frequency of all five extreme events in the same year; then, MXCI was transformed into the composite risk grade index determined by $CF_{\mathrm{MXCI}}$ in the nationwide comparison, with 4, 3, 2, and 1 denoting very high risk, high risk, medium risk, and low risk, respectively.

### 2.9. Classification Thresholds for the Risk Levels of Extreme Climate Events in China

Table 2 shows the annual average frequency of extreme high-temperature, extreme low-temperature, extreme drought, extreme precipitation, and extreme typhoon events in China from 1991 to 2020, the thresholds for the risk levels determined by the percentile method, and the level thresholds for the composite risk grade index of multi extreme events. In this study, the classification threshold of single extreme event and the classification thresholds for composite risk grade of multi extreme events in China are subject to Table 2.

**Table 2.** Risk grades of different extreme events in China in the 30 years from 1991 to 2020.

| Name | Abbreviation | 4 (Very High Risk) | 3 (High Risk) | 2 (Medium Risk) | 4 (Low Risk) |
|---|---|---|---|---|---|
| Risk for extreme high-temperature events | $R_{ht}$ | ≥13.52% | <13.52%, ≥12.59% | <12.59%, ≥11.86% | <11.86% |
| Risk for extreme low-temperature events | $R_{lt}$ | ≥8.82% | <8.82%, ≥7.95% | <7.95%, ≥6.93% | <6.93% |
| Risk for extreme drought events | $R_{dr}$ | ≥8.52% | <8.52%, ≥7.32% | <7.32%, ≥6.32% | <6.32% |
| Risk for extreme precipitation events | $R_{pr}$ | ≥6.58% | <6.58%, ≥5.03% | <5.03%, ≥2.82% | <2.82% |
| Risk for extreme typhoon events | $R_{ty}$ | ≥0.64% | <0.64%, ≥0.35% | <0.35%, ≥0.13% | <0.13% |
| Composite risk for multi extreme events | MXCI | ≥13.00% | <13.00%, ≥11.00% | <11.00%, ≥8.00% | <8.00% |

### 2.10. Validation

To verify the validity of the MXCI index, we compared it with the Climate Risk Index (CRI) established by WANG (Figure 1). We selected Hangzhou Station in Zhejiang Province in southeast China (Figure 1a), Dali Station in Yunnan Province in southwest China (Figure 1b), Changchun Station in Jilin Province in northeast China (Figure 1c), and Luochuan Station in Shaanxi Province in northwest China (Figure 1d), respectively, to give the annual change curves and linear trends of MXCI and CRI from 1961 to 2020.

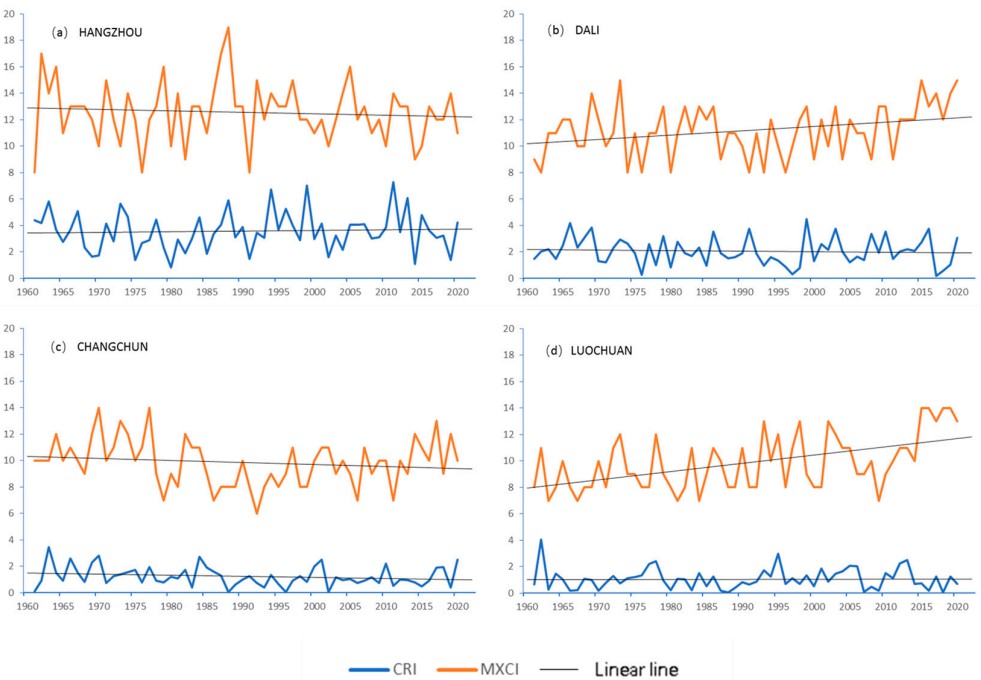

**Figure 1.** Annual change curves and linear trend (grey line) of MXCI (orange line) and CRI (blue line) in Hangzhou (**a**), Dali (**b**), Changchun (**c**) and Luochuan (**d**) station from 1961 to 2020.

The correlation coefficients of Hangzhou, Dali, Changchun, and Luochuan were 0.36, 0.32, 0.36, and 0.18, respectively. The correlation coefficients of other stations except Luochuan passed the significance test of 0.05. Compared with the 60-year change characteristics of the four stations, the linear trend of the two indexes was basically the same, but the interannual and interdecadal change characteristics of the MXCI were more obvious; especially, the MXCI of Dali, Changchun and Luochuan stations had shown a significant increasing trend since the 1990s, but the change characteristics of the CRI were not obvious. Analyzing the reasons, it was related to the use of disaster loss as a weight factor in the construction of CRI. As the collection of disaster data was limited by natural and man-made factors, the CRI index cannot reflect the climate extremes well. From the above comparison and acceptance, MXCI reflects the extreme climate better than the CRI.

*2.11. Data and Regionalization*

In this study, the meteorological data from 1961 to 2020 are obtained from 2254 meteorological stations in China, including daily precipitation, mean air temperature, maximum air temperature, and minimum air temperature, which are provided by the National Meteorological Information Center of the China Meteorological Administration. Note that the homogenization test and correction have been performed on the mean air temperature, maximum air temperature, and minimum air temperature [42,43]. The typhoon data consist of number, name, day, month and year of occurrence, intensity level every 6 h, central position (longitude and latitude), central pressure and wind speed, etc., which are obtained from the best track database of tropical cyclones from 1949 to 2020 provided by the Shanghai Typhoon Institute of China Meteorological Administration (https://tcdata.typhoon.org.cn/zjljsjj_zlhq.html (accessed on 12 May 2022)).

In this study, the average value of 1991–2020 is used to represent the climate state. The linear trend analysis is performed on the data from 1961 to 2020, and the calculated linear trend is tested for significance by using the Mann–Kendall test method [44]. In order to analyze the variation characteristics of regional extreme events, China is divided into seven regions (as shown in Figure 2), which is mainly based on the rules of geographical location and climate similarity. Specifically, the seven regions are northwest China (include

5 provinces, i.e., Xinjiang, Gansu, Qinghai, Ningxia, and Shaanxi), north China (include 5 provinces or municipalities, i.e., Beijing, Tianjin, Hebei, Shanxi, and Inner Mongolia), northeast China (include 3 provinces, i.e., Liaoning, Jilin, and Heilongjiang), east China (include 7 provinces or municipalities, i.e., Shanghai, Jiangsu, Anhui, Shandong, Zhejiang, Jiangxi, and Fujian), central China (include 3 provinces, i.e., Henan, Hubei, and Hunan), south China (include 3 provinces, i.e., Guangdong, Guangxi, and Hainan), and southwest China (include 5 provinces or municipalities, i.e., Sichuan, Chongqing, Guizhou, Yunnan, and Tibet).

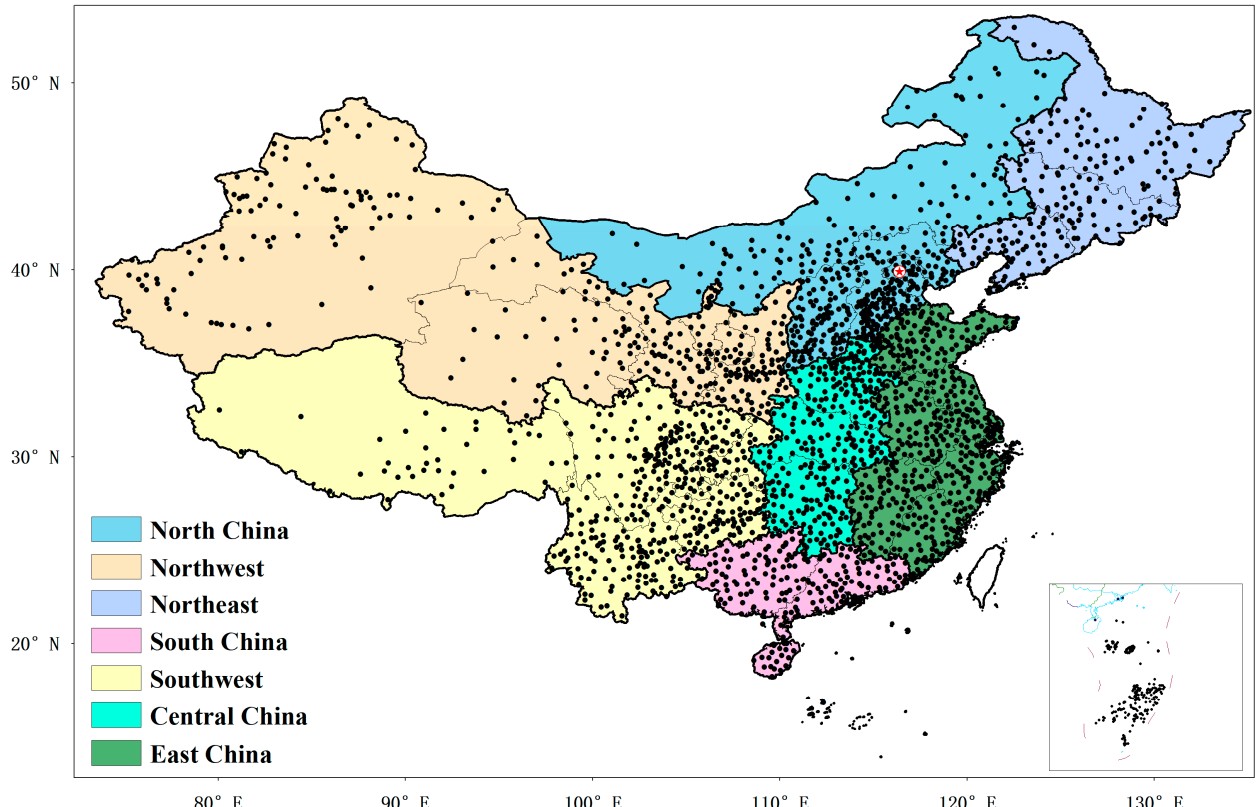

**Figure 2.** Distribution of meteorological stations (black point) and regional division in China (different colors represent different divisions, the red star represents the location of Beijing).

## 3. Result

### 3.1. Extreme High Temperature and Low-Temperature Events

With global warming, extreme high-temperature events have occurred frequently in China in the past 30 years (1991–2020) (Figure 3a), with an annual average frequency of 12.19% (Table 3). In terms of geographical distribution, extreme high-temperature events mainly occurred in western China and south China. From the perspective of the linear trend in 1961–2020 (Figure 3b), most areas of China have shown an increasing trend of extreme high-temperature events, especially in most of western and northern China, where the increase is more obvious. The increasing trend in most regions of China has passed the significance test at the 99% confidence level (Table 4), indicating that the extreme high-temperature events in China have increased significantly in the past 60 years owing to global warming.

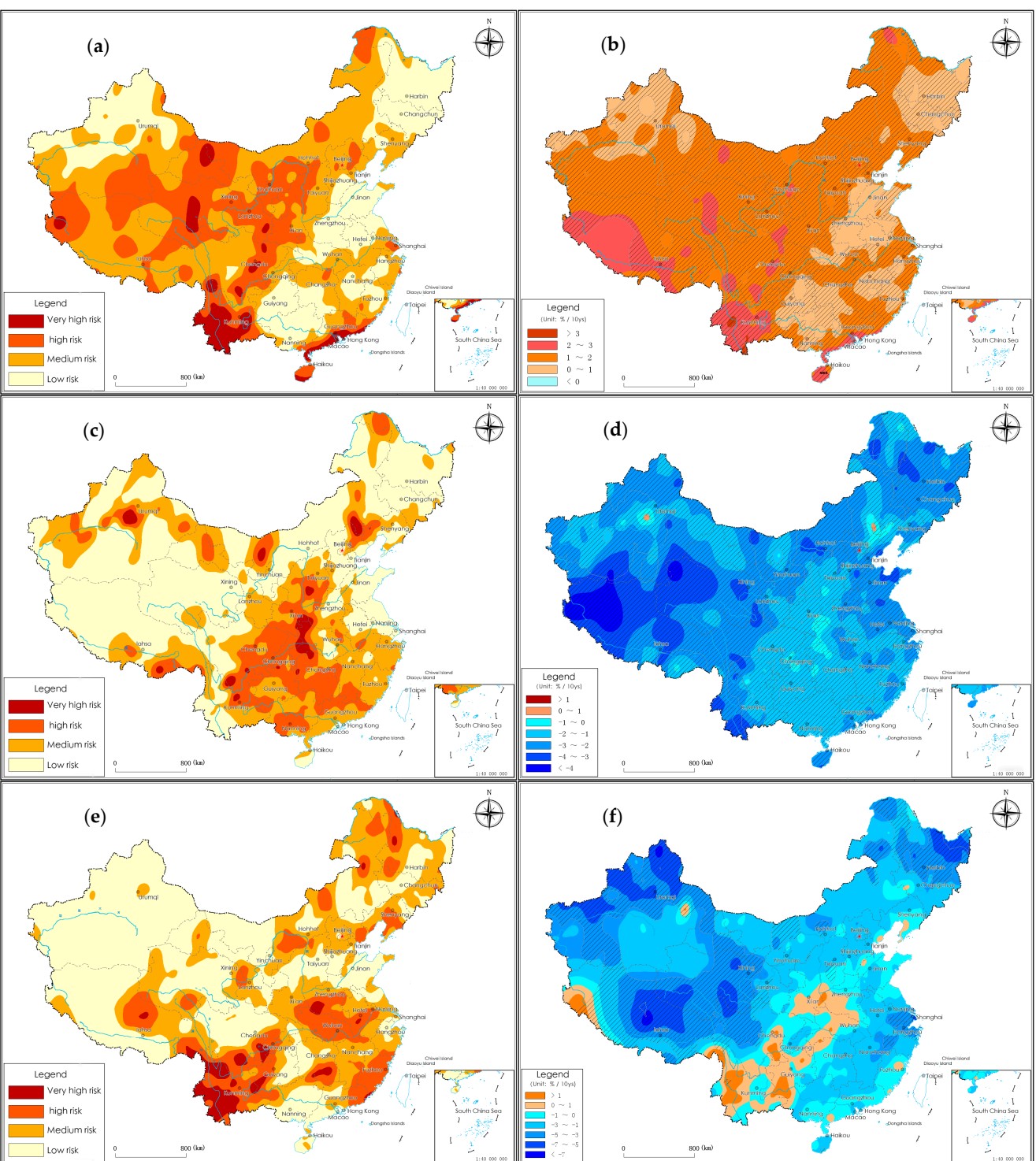

**Figure 3.** *Cont.*

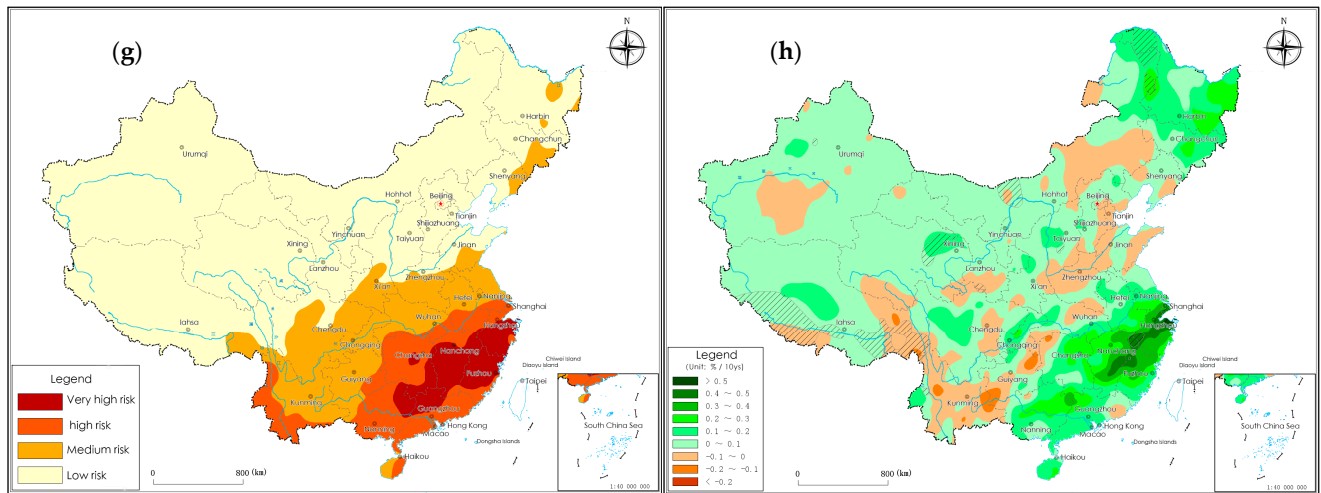

**Figure 3.** Risk of different extreme events in China in the 30 years from 1991 to 2020 (extreme high-temperature events (**a**), extreme low-temperature events (**c**), extreme drought events (**e**), and extreme precipitation events (**g**) and the linear trend of annual frequency of extreme events in the 60 years from 1961 to 2020 (extreme high-temperature events (**b**), extreme low-temperature events (**d**), extreme drought events (**f**), and extreme precipitation events (**h**); the tilt-line-covered areas indicate statistically significant test pass at the 95% confidence level. Unit is %/10a in (**b**,**d**,**f**,**h**)).

**Table 3.** The annual average frequency of different extreme events in the 30 years from 1991 to 2020 in different regions of China.

| Name | Northeast China | North China | Northwest China | East China | Central China | South China | Southwest China | Whole China |
|---|---|---|---|---|---|---|---|---|
| Extreme high-temperature events | 11.63 | 12.05 | 12.49 | 11.90 | 11.70 | 12.82 | 12.64 | 12.19 |
| Extreme low-temperature events | 6.55 | 6.81 | 7.25 | 6.88 | 7.55 | 7.56 | 7.70 | 7.19 |
| Extreme drought events | 6.73 | 6.45 | 5.94 | 6.78 | 7.30 | 6.19 | 7.23 | 6.70 |
| Extreme precipitation events | 2.65 | 2.04 | 1.50 | 5.38 | 4.79 | 6.01 | 4.12 | 3.77 |
| Extreme typhoon wind speed events | 0.00 | 0.00 | 0.00 | 0.08 | 0.02 | 0.21 | 0.00 | 0.04 |
| Extreme typhoon precipitation events | 0.00 | 0.00 | 0.00 | 0.14 | 0.04 | 0.34 | 0.00 | 0.07 |

**Table 4.** The linear trend of annual frequency of different extreme events in the 60 years from 1961 to 2020 in different regions of China. (Unit: %/10 a, '**' represent statistically significant test pass at the 99% confidence level; '*' means pass the 95% level).

| Name | Northeast China | North China | Northwest China | East China | Central China | South China | Southwest China | Whole China |
|---|---|---|---|---|---|---|---|---|
| Extreme high-temperature events | 0.930 ** | 1.540 ** | 1.400 ** | 1.010 ** | 0.840 ** | 1.590 ** | 1.600 ** | 1.220 ** |
| Extreme low-temperature events | −2.270 ** | −2.270 ** | −1.970 ** | −2.140 ** | −1.740 ** | −1.840 ** | −1.720 ** | −1.980 ** |
| Extreme drought events | −0.890 ** | −0.500 * | −1.070 ** | −0.920 ** | −0.410 | −0.450 | −0.160 | −0.670 ** |
| Extreme precipitation events | 0.110 | 0.030 ** | 0.040 | 0.200 * | 0.080 | 0.140 | 0.000 | 0.070 * |
| Extreme typhoon wind speed events | 0.000 | 0.000 | 0.000 | −0.042 ** | −0.021 ** | −0.118 ** | 0.000 | −0.023 ** |
| Extreme typhoon precipitation events | 0.000 | 0.000 | 0.000 | 0.008 | −0.001 | −0.008 | 0.000 | 0.001 |

Many researchers [45–48] have used 35 °C as the high-temperature threshold to study the extreme high-temperature events in China. The results showed that the frequency of extreme high-temperature events in most regions of China shows an increasing trend, which is consistent with the results of this study. The above analysis shows that with global warming the high frequency of extreme high-temperature events in China still shows continuity in recent years, characterized by long duration, high intensity, and wide impact range. In addition, the extreme high-temperature events have emerged in spring and autumn, which deserves attention.

In the past 30 years, extreme low-temperature events mainly occurred in the eastern part of northwest China and the south of the Yangtze River, of which southwest China, south China, and central China are the areas with frequent extreme low-temperature events, while most of northern and western China has relatively low frequency of extreme low-temperature events (Figure 3c). Since 1961, extreme low-temperature events in China have shown a remarkable decreasing trend with global warming (Figure 3d). The nationwide annual frequency of extreme low-temperature events decreases by 1.98% per decade on average (Table 4), with the decreasing trend in most regions passing the significance test at the 99% confidence level.

From the perspective of the occurrence in this century, the frequency of extreme low-temperature events shows a decreasing trend, but the intensity is undiminished. On the contrary, the intensity and impact range of some extreme low-temperature events and the losses are historically rare. The researches show that [6] there were 29 cold air processes affecting China in 2021, including 11 cold wave processes, which is obviously more than normal (5.2 annual average). The cold wave processes in early January and early November of 2021 were characterized by a strong cooling range and serious impacts. Low-temperature freezing and snow disasters occurred in many places of northern China, and some southern provinces also experienced snow and freezing rain, which has a great impact on agricultural and animal husbandry production, traffic transportation, energy supply, as well as the lives of residents. Strong cooling is accompanied by heavy snowfall, gales, and other severe weather, and the impacts of such compound extreme event are extremely serious. Some researchers have studied cold nights and frost days [49,50] and found that the frequency of extreme low-temperature events in China has shown a significant decreasing trend, but there has been an increasing trend of extreme low-temperature events since 2007. Accordingly, in the context of global warming, special attention needs to be paid to the monitoring and early warning of extreme low-temperature events.

### 3.2. Extreme Drought Events

Over the past 30 years, extreme drought events mainly occurred in central and southern China (Figure 3e), with the highest annual frequency of 7.30% and 7.23% in central China and southwest China, respectively (Table 3). Specifically, the annual frequency of extreme drought events reaches about 10% in most of Yunnan, southeastern Sichuan, western Guizhou, northern Hubei, southwestern Henan, eastern Guangdong, and western Fujian. The frequency of extreme drought events is also relatively high in parts of northeast China and north China, but the occurrence areas are relatively scattered.

The nationwide extreme drought events have shown a decreasing trend since 1961 (Figure 3f), with the annual frequency decreased by 0.67% per decade on average (Table 4). The areas with the most obvious reduction are mainly located in the three northern regions (northeast China, north China, and northwest China): the Qinghai–Tibet Plateau and adjacent regions and most of southeastern China. Specifically, the decreasing trend is the most evident in most of Heilongjiang, northern Xinjiang, and central Tibet with a reduction of more than 5.0% per decade on average, which has passed the significance test at the 99% confidence level. The areas with increased extreme drought events are mainly located in the drought-prone belt from southwest China to southern northeast China. Specifically, the increasing rate of extreme drought events is about 1.0% per decade on average in

northwestern Yunnan, western Guizhou, western Guangxi, southwestern Sichuan, southern Shaanxi, north–central Hubei, and western Henan but is not statistically significant.

From the above analysis, with the influence of global warming, there is an increasing trend in precipitation in most regions of China, which leads to a decreasing trend in extreme drought events. However, due to the uneven spatio–temporal distribution of precipitation and the increase in evapotranspiration caused by temperature rise, regional and periodic droughts are evident in China. There were many studies [15,51–53] on the characteristics of drought climate change in China in recent decades, and the results showed that there is a clear warm–wet trend in northwest China and the areas adjacent to the Qinghai–Tibet Plateau, where the frequency of droughts is reduced. Drought events increase in southwest China (excluding the Tibet Autonomous Region). These are generally consistent with the results of this study. Nevertheless, many studies [54–57] showed that there is an increasing trend in drought events in northeast China, north China, and the middle and lower reaches of the Yangtze River and south China, which contradicts the results of our study. There are three reasons for this discrepancy. Firstly, the study period is different, with data dating to around 2010 in most studies but updating to 2020 in this study. Secondly, the drought indexes used for study are different, and different factors considered for different indexes can lead to different results. Thirdly, this study focuses on extreme drought events, while most scholars focus on all drought events.

### 3.3. Extreme Precipitation Events

In the past 30 years, extreme precipitation events mainly occurred in the southern region of China (Figure 3g), with the highest annual average frequency in south China (6.01%) and east China (5.38%), followed by central China (4.79%) and southwest China (4.12%) (Table 3). There are relatively few extreme precipitation events in northern China, with an annual average frequency of 2–3% in northeast China and north China. Due to less precipitation in northwest China, there are also few extreme precipitation events with a frequency of only 1.5%.

Over the past 60 years, extreme precipitation events tended to increase in most of China (Figure 3h), with the nationwide annual frequency increased by 0.07% per decade on average, which has passed the significance test at the 95% confidence level (Table 4). The most apparent increase in extreme precipitation events occurs in north China and east China, with an average increase of 0.03% and 0.2% per decade, respectively. The increase in north China and east China has passed the significance test at the 99% and 95% confidence levels, respectively. Extreme precipitation events show an increasing trend in south China, northeast China, central China, and northwest China, but the increase fails to pass the significance test. Extreme precipitation events tend to decrease in some areas of southwest China, central China and north China, which is generally consistent with the regions where extreme drought events have increased, but the decrease in most areas fails to pass the significance test.

The researches show that [6,9] there 36 regional rainstorm processes occurred in China in 2021. Especially, a record-breaking severe rainstorm hit Henan Province and caused great property loss and casualties. On 17–24 July 2021, an extreme precipitation event hit many parts of Henan; the daily precipitation in 19 counties and cities including Zhengzhou has surpassed the historical records, and the precipitation for 3 consecutive days of 32 counties and cities breaks the historical records. The maximum hourly rainfall in Zhengzhou (201.9 mm/h) surpassed the unprecedented heavy rainfall on August 4–8, 1975 in Linzhuang (198.5 mm/h) in Henan province, setting a new record of hourly rainfall in Chinese mainland.

From the above analysis, it is clear that in the context of global warming, there is an increasing trend in extreme precipitation in most parts of China and this trend is relatively significant. Most Chinese researchers used 50 mm daily precipitation as the threshold to study extreme precipitation events [6,9,40], while the results are consistent with this study. Several researchers also studied the contribution of extreme precipitation to total

precipitation, as well as the intensity and duration of extreme precipitation [58], and the results showed that the intensity and duration of extreme precipitation in southern China have increased in recent decades. The IPCC Sixth Assessment Report [1] pointed out that for every 1 °C increase of global temperature atmospheric water vapor can increase by approximately 7%. As the climate warms, the atmosphere can hold more water vapor before it becomes saturated, making it easier to form extreme precipitation processes. The combination of abundant water vapor conditions and special atmospheric circulation can lead to extreme precipitation event; this is the main reason for the increase of extreme precipitation events in China over recent decades. The increase in extreme precipitation events brings new challenges to regional water resource management. Furthermore, urban floods and many geological disasters are closely linked to extreme precipitation. Thus, more research is needed on this aspect to develop effective preventive and counter measures.

### 3.4. Extreme Typhoon Events

Extreme typhoon events include both extreme typhoon wind speed (Figure 4a) and extreme typhoon precipitation (Figure 4c). Extreme typhoon events are mainly distributed in south China, southeastern coastal areas, and parts of central China, so in this study we focus on extreme typhoon events in south China, east China, and central China. Over the past 30 years, the most extreme typhoon events occurred in south China, where the annual average frequency of extreme typhoon precipitation and extreme typhoon wind speed are 0.34% and 0.21%, respectively (Table 3). The corresponding values in some coastal areas reach 0.5% and 0.4%, respectively, indicating that typhoons have the greatest impact on the coastal areas of South China. The second highest frequency occurred in east China, with an annual average frequency of 0.14% and 0.08% for extreme typhoon precipitation and extreme typhoon wind speed events, respectively, and the impact areas are mainly located in the southern coastal areas of Zhejiang Province and Fujian Province. Extreme typhoon events are less frequent in central China, with an annual average frequency of 0.04% and 0.02% for extreme typhoon precipitation and extreme typhoon wind speed events, respectively, and the impact areas are mainly located in the southern parts of Hunan Province and Jiangxi Province.

Since 1961, extreme typhoon wind speed events present a decreasing trend nationwide (Figure 4b), with the most obvious decrease in south China, where the annual frequency is decreased by 0.118% per decade on average, and by 0.042% and 0.021% in east China and central China, respectively (Table 4). All the trends have passed the significance test at the 99% confidence level, indicating that the decrease in extreme typhoon wind speed in China is relatively significant owing to the influence of global warming. Changes in extreme typhoon precipitation since 1961 (Figure 4d) are not exactly the same as the changes in extreme typhoon wind speed. Extreme typhoon precipitation events in south China (excluding Hainan Province) and central China are on a decreasing trend, approximately ranging from −0.01% to −0.03%. However, in most of east China and Hainan Province, they are on an increasing trend, ranging from 0.01% to 0.04%, with a pronounced increase in the coast of Zhejiang Province and northeast Hainan. Nationally, both the decrease and increase in extreme typhoon precipitation do not pass the significance test at the 95% confidence level, indicating that the linear trend of extreme typhoon precipitation is nonsignificant.

Related studies [6,59–61] have shown that the number of typhoons generated in the northwest Pacific and the South China Sea showed a decreasing trend in recent decades, with the abrupt change in 1995, accompanied by the decreased typhoon annual maximum wind speed and annual precipitation. But, the typhoon-induced precipitation in the southeast coastal areas was on an increasing trend [62], which is basically consistent with the results of this study. What should be paid attention to is that in recent years, the number of northbound typhoons affecting China is increasing, which has a great impact on the eastern coastal areas, include middle and high latitudes, especially the impact of extreme typhoon precipitation [11–13,63,64].

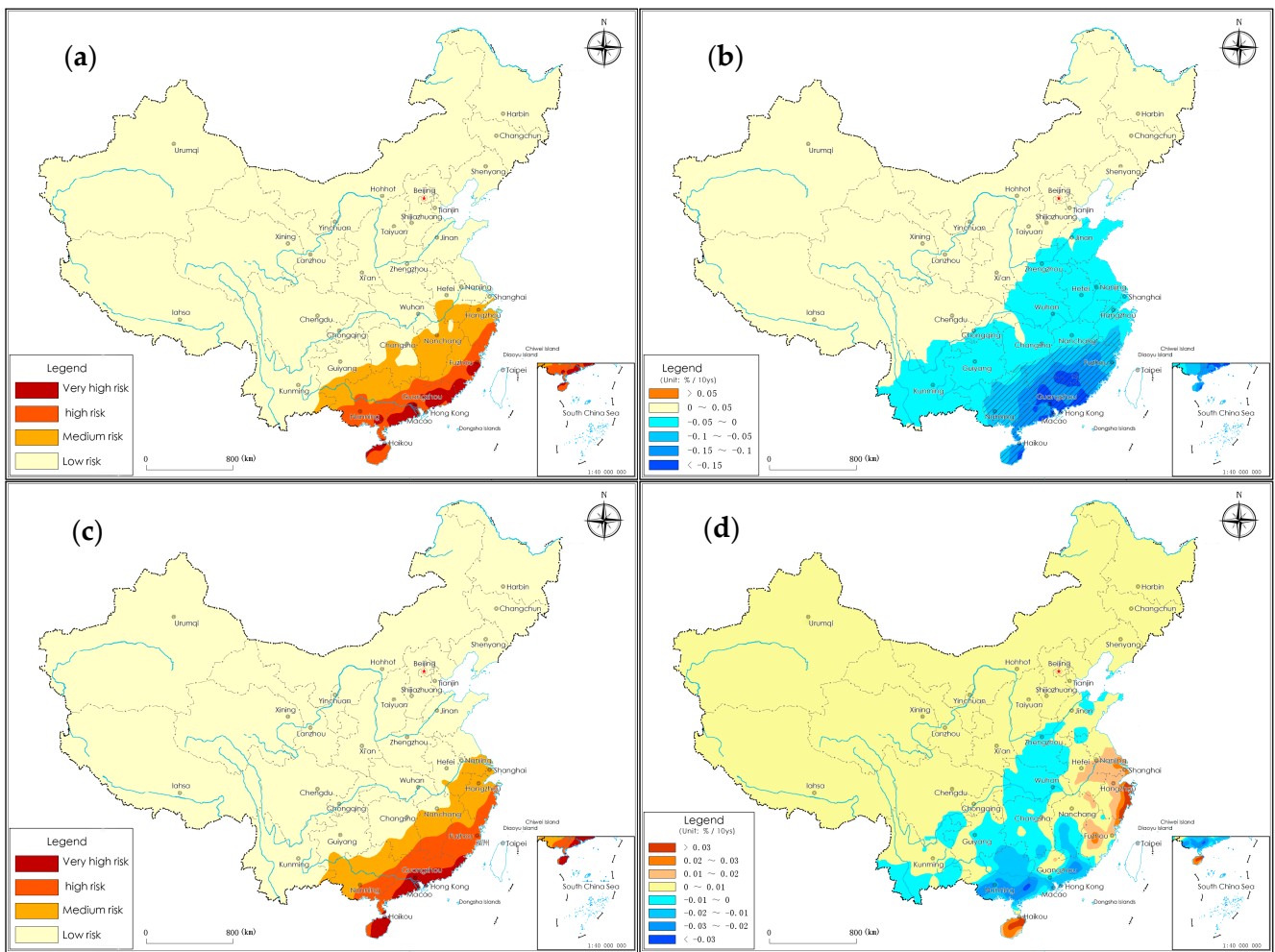

**Figure 4.** Risk of extreme typhoon events in China in the 30 years from 1991 to 2020 (extreme typhoon wind speed events (**a**) and extreme typhoon precipitation (**c**)) and the linear trend of annual frequency of extreme typhoon events in the 60 years from 1961 to 2020 (extreme typhoon wind speed events (**b**) and extreme typhoon precipitation (**d**); the tilt-line-covered areas indicate statistically significant test pass at the 95% confidence level. Unit is %/10a in (**b**,**d**)).

### 3.5. The Composite Grade Index of Extreme Climate Events

The extreme high temperature, extreme low temperature, extreme drought, extreme precipitation, and extreme typhoon events are compared nationwide to determine their risk levels, and then the risk levels of individual extreme events are composited to obtain the multi extreme events composite risk grade index (MXCI). The analysis of MXCI allows us to grasp the comprehensive characteristics and trends of extreme climate in different regions of China. Extreme climate events during 1991–2020 mostly occurred in central and southern China (Figure 5a), and the areas with high risk levels of MXCI are mainly located in south China, the southeast coastal areas, and southern central China (Table 5). The MXCI in Guangdong, Fujian, southeastern Zhejiang, southern Jiangxi, south–central Hunan, and western Guangxi are at high or very high risk levels, indicating that these regions are highly affected by extreme climate events. Southwest China, such as Yunnan and southeastern Sichuan, also has relatively high MXCI, indicating that southwest China is also significantly affected by extreme climate.

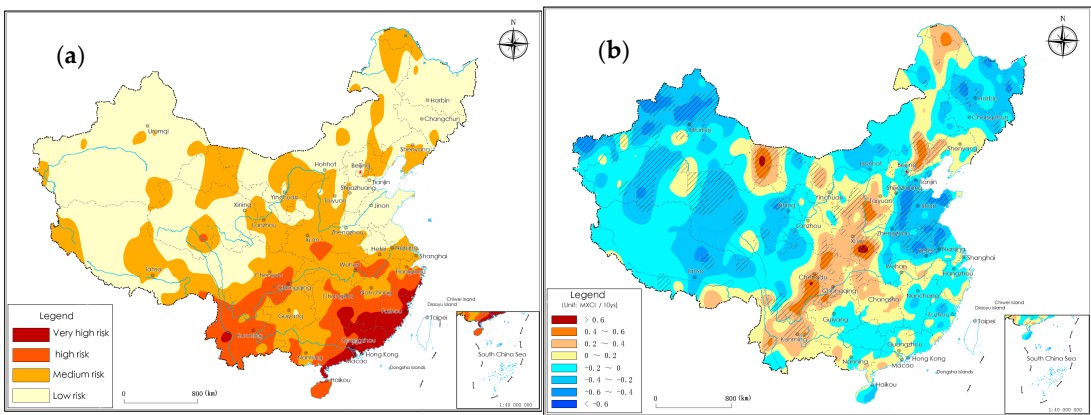

**Figure 5.** Composite risk of multiple extreme climate events (annual average of MXCI in China in the 30 years from 1991 to 2020 (**a**); the linear trend of MXCI in the 60 years from 1961 to 2020 (**b**). The tilt-line-covered areas indicate statistically significant test pass at the 95% confidence level. Unit is %/10a in (**b**)).

**Table 5.** The annual average of MXCI in the 30 years from 1991 to 2020 and the linear trend of MXCI in the 60 years from 1961 to 2020 in different regions of China. ('**' represent statistically significant test pass at the 99% confidence level).

| Name | Northeast China | North China | Northwest China | East China | Central China | South China | Southwest China |
|---|---|---|---|---|---|---|---|
| Annual average | 1.82 | 1.73 | 1.79 | 2.48 | 2.37 | 3.05 | 2.35 |
| Linear trend | −0.03 | −0.03 | 0.02 | −0.06 ** | −0.01 | 0.03 | 0.09 ** |

By analyzing the linear trend of MXCI in China since 1961 (Figure 5b), the study area can be roughly divided into three zones. Zone 1 refers to the western region, including northwest China, the areas adjacent to the Qinghai–Tibet Plateau, and west–central Inner Mongolia. Zone 2 is the eastern region, including south China, east China, central China, southern–central north China, and eastern–central northeast China. Zone 3 is the middle region, i.e., the belt zone from Southwest China to Northeast China, including eastern–central southwest China, eastern northwest China, western central China, and northern north China. The MXCI in most areas of Zone 1 and Zone 2 shows a decreasing trend, among which the index in most of Xinjiang, the areas adjacent to the Qinghai–Tibet Plateau, central Inner Mongolia, southern–central north China, the Huanghuai region, most of northeast China, and parts of southeast coastal regions decreases remarkably, and the linear trend of some regions has passed the significance test at the 95% confidence level. The MXCI in Zone 3 shows an increasing trend, with remarkably increase in northern–central Yunnan, eastern and southern Sichuan Basin, southern Shaanxi, southern Shanxi, northern Hebei, and western Liaoning. Almost all the increasing trends have passed the significance test at the 95% confidence level.

From the above analysis it can be seen that in recent decades the high-incidence areas of extreme climate events are mostly located in south and central China, especially in south China where multi extreme climate events are concentrated. From the linear trend of MXCI over the past 60 years, the regions with an obvious increase in the risk levels of extreme events are mainly located in the belt from southwest China to the southern northeast China, as a result of the frequent occurrence of extreme high temperature, extreme drought, and extreme low-temperature events in these regions in recent decades. This result indicates that these regions are prone to compound extreme climate events, which is basically consistent with Chen et al. and Wang et al. [65,66]. The MXCI presents a decreasing trend in most of the western, northern, and eastern China. This is mainly due to the increase in precipitation

in these regions in recent decades as a result of climate warming, which has led to an obvious reduction in extreme drought and extreme low-temperature events, as well as a prominent reduction in extreme typhoon events in southeastern China.

## 4. Discussion

The number of extreme climate events is not only closely linked to climate change but also to the criteria for identifying extreme events. For instance, the determination of extreme precipitation events in this study not only considers the extremity of precipitation for 3 consecutive days but also limits the magnitude of precipitation; i.e., the amount of precipitation for 3 days must exceed 30 mm, thus excluding the influence of extreme precipitation events in arid areas and focusing on extreme precipitation events with large precipitation. The 90-day SPI index is used here to study extreme drought events, but the SPI with other time scale or other drought index (such as SPEI) can also be utilized to study extreme drought events. When studying extreme high-temperature events, if the focus is on the effects of heat waves on human health, it is recommended to add absolute thresholds (e.g., the daily maximum temperature must be greater than 30 °C) as limited conditions. The limited conditions of minimum temperature (e.g., daily minimum temperature $\leq 0$ °C) can also be added when studying extreme low-temperature events. When we study extreme typhoon events, we only consider the impact range of 500 km around the eye of the typhoon, but sometimes the impact of the typhoon extends far beyond this range. Accordingly, the criteria for identifying extreme weather events are important, and appropriate indexes should be carefully selected based on the study objective and area.

In the past 30 years, the high-value areas of MXCI are mainly distributed in south China and the southeast coastal area, while the northern and western parts of China have an obviously low MXCI, mainly due to the impact of extreme typhoon events. This also indicates that the MXCI reflects not only the frequency of individual extreme events but also the number of types of extreme events. If a region has a multitype and high frequency of extreme events, the risk level reflected by the MXCI will be higher. In this paper, five kinds of extreme climate events are selected to construct the MXCI index for the Chinese region. However, China has a large area, and different regional climate characteristics and types of extreme climate events are different. For example, the northwest and north China are rarely affected by typhoons, which leads to a low MXCI index in these regions. Therefore, MXCI research should be carried out in regions with similar climate and extreme climate events.

The extreme climate events in this study only involve the five types that have a great impact nationwide, and other extreme events such as gale, hail, sandstorm, and tornado are not considered here. Thus, the MXCI in this study only reflects part of the characteristics of extreme climate in China. With regard to how many extreme events should be selected to reflect the overall characteristics of extreme events in a region, the analysis suggests that to fully grasp the overall characteristics of extreme climate events in the region it is necessary to select the events with high extremity, frequent occurrence, and severe impacts.

In the summer of 2022, a severe high-temperature drought event occurred in the Yangtze River Basin [10], causing serious impacts on local industrial and agricultural production. Less precipitation compounded by persistent high-temperature weather can easily trigger a compound drought, which is a new trend in recent years [67–70]. However, the MXCI does not take into account this compound extreme events, which is an area where the MXCI needs to be improved in the future.

In this study, only the statistical characteristics and comprehensive risk of five extreme climate events are analyzed, and the impact objects and loss caused by extreme events are not taken into account. In order to effectively respond to the impacts and risks of extreme climate events, an overall consideration of the occurrence time, intensity, impact area, impact objects, and loss of each extreme event should be considered.

## 5. Conclusions

Over the past 60 years, extreme high-temperature events in China have shown a significant increasing trend, especially in most of western and northern China. The average proportion of extreme high-temperature events per decade increased by 1.22% (about 4.5 times). In recent 30 years, extreme high temperature events have occurred frequently in China, with an average annual daily frequency of 12.19% (about 44.5 times), and the high incidence areas are mainly located in western China and southern China. In the last 60 years, extreme low-temperature events in China have shown a remarkable decreasing trend, with the most obvious decrease in northern China and the Qinghai–Tibet Plateau and its adjacent areas. The average proportion of extreme low-temperature events per decade decreased by 1.98% (about 7.2 times). Over the past 30 years, extreme low-temperature events mainly occurred in central and southern China, of which the eastern Sichuan Basin, Chongqing, and Guizhou are the areas with frequent extreme low-temperature events.

The precipitation in most regions of China has shown an increasing trend over the past 60 years, which leads to a decreasing trend in extreme drought events. The average proportion of extreme drought events per decade decreased by 0.67% (about 2.4 times). The areas with the most obvious reduction are mainly located in northern China, the Qinghai–Tibet Plateau and adjacent areas, and most of southeastern China. The area of increasing extreme drought events is mainly located from southwest to the south northeast of China. In the past 30 years, extreme drought events mainly occurred in southwest China, central China, and northeast China, with an average annual frequency of 6.7% (about 24.4 times) in the whole of China.

Extreme precipitation events mainly occur in southern China in the past 30 years, with the highest annual average frequency in south China (6.01%, about 21.9 times) and east China (5.38%, about 19.6 times). With global warming, extreme precipitation has shown an increasing trend in most regions of China in the past 60 years, especially in north China and east China with the most remarkable increases. The average proportion of extreme precipitation events per decade increased by 0.07% (about 0.3 times), meanwhile 0.03% (about 0.1 times) and 0.2% (about 0.7 times) in north China and east China, respectively.

Extreme typhoon events are distributed in south China, southeastern coastal areas, and parts of central China. Over the past 30 years, the most extreme typhoon events occurred in south China, where the annual frequency of extreme typhoon events in some coastal areas reached 4–5% (about 14–18 times). In the last 60 years, extreme typhoon wind speed events in China presented a significant decreasing trend. Extreme typhoon precipitation events decreased in south China but increased in east China, though not significantly.

The analysis of MXCI in the past 30 years shows that the regions with higher risk levels are mainly located in central and southern China, with the highest in south China, the southeastern coastal region and southern central China, followed by southwest China. Over the past 60 years, the MXCI has shown a decreasing trend in western China and most of southeastern China, and an increasing trend in the middle zone from southwest China to northeast China, with obvious increases in northern–central Yunnan, eastern and southern Sichuan Basin, southern Shaanxi, southern Shanxi, northern Hebei, and western Liaoning. High priority attention should be given to the trend of more types and frequency of extreme weather and climate events, as well as their impacts and risks on social and economic development.

**Author Contributions:** Conceptualization, C.Z., C.X. and S.L.; Data curation, C.Z., X.C. and Z.S.; Formal analysis, C.Z., Y.R. and S.Z.; Investigation, C.Z. and C.X.; Methodology, C.Z.; Resources, C.Z. and Y.R.; Software, C.Z., C.X., S.L., Y.R., X.C. and Z.S.; Validation, C.Z., C.X., S.L. and Z.S.; Visualization, C.Z. and S.Z.; Writing—original draft, C.Z., S.Z. and Z.S.; Writing—review and editing, C.Z., C.X., S.L., Y.R., S.Z. and X.C. All authors have read and agreed to the published version of the manuscript.

**Funding:** This research was funded by the National Key Research and Development Program of China (2020YFA0608203), China Three Gorges Corporation (No. 0704181) and The National Natural Science Foundation of China (No.52109024).

**Data Availability Statement:** The data analyzed in this study are subject to the following licenses/restrictions: The dataset can only be accessed from inside China Meteorological Administration. Requests to access these datasets should be directed to Cunjie Zhang, zhangcj@cma.gov.cn.

**Acknowledgments:** In the process of revising this paper, the authors have received strong guidance from Ren Guoyu of CMA National Climate Centre, and Ren Fumin of Chinese Academy of Meteorological Sciences. The authors are very grateful to the comments of distinguished editor-in-chief and peer reviewers to improve the quality of this article.

**Conflicts of Interest:** The authors declare no conflict of interest.

## Abbreviations

| | |
|---|---|
| 6.1 | Ten-year average (10 a) |
| 6.2 | Cumulative frequency (CF) |
| 6.3 | Extreme high-temperature event ($E_{th}$) |
| 6.4 | Extreme low-temperature event ($E_{tl}$) |
| 6.5 | Standardized Precipitation Index (SPI) |
| 6.6 | Extreme drought event ($E_{dr}$) |
| 6.7 | Precipitation of 3 consecutive days (R3) |
| 6.8 | Extreme precipitation event ($E_{pr}$) |
| 6.9 | Extreme typhoon event ($E_{ty}$) |
| 6.10 | Extreme daily typhoon precipitation event ($E_{tp}$) |
| 6.11 | Extreme daily typhoon maximum wind speed event ($E_{tw}$) |
| 6.12 | Risk for extreme high-temperature events ($R_{ht}$) |
| 6.13 | Risk for extreme low-temperature events ($R_{lt}$) |
| 6.14 | Risk for extreme drought events ($R_{dr}$) |
| 6.15 | Risk for extreme precipitation events ($R_{pr}$) |
| 6.16 | Risk for extreme typhoon events ($R_{th}$) |
| 6.17 | Multi extreme events composite risk grade index (MXCI) |

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
