# Peer review of "Analysis of the Composite Risk Grade for Multi Extreme Climate Events in China in Recent 60 Years"

_climate, doi:10.3390/cli11090191_

Round 1

Reviewer 1 Report

Comments are provided in review report.

Reviewer 2 Report

The manuscript entitled "Analysis of the Composite Risk Grade for Multi Extreme Climate Events in China in Recent 60 Years" is a major improvement on the subject of another article published in the Chinese Journal of Geophysics-Chinese Edition (DOI: 10.6038/cjg2022Q0255, in Chinese). I recommend authors to add it to the bibliography.

The multi extreme events composite risk grade index (MXCI) represents an attempt to objectively reflect the comprehensive characteristics of multi extreme climate events in different regions. This statement of the authors should be more reserved, in general these indicators are relevant mostly for comparisons, and not necessarily for relevance. Relevance is something relative that can be improved over time, there is no such thing as absolute objectivity. But I agree that it is a better composite indicator than other known ones. 

This manuscript is a timely contribution to the field.

Conclusions are presented in an appropriate fashion and are well supported by the data.

I agree with the publication in this journal, but suggest the following minor revisions to the author to improve the presentation of the research results. These are only for an easier understanding by the readers of the research results.

Overall, the study is methodologically sound with promising results. There are some comments to be addressed.

Minor Revisions:

Comment 1: Similarities with the uncited article should be avoided, especially since it represents an advanced research with a different proposal than the multi extreme events composite risk grade index (MXCI). The rewriting must be limited to the case study, where I saw similarities. It needs little elaboration;

Comment 2: The cumulative distribution must be specified. I suspect it is the most used, namely the two-parameter Gamma, and for the thresholds where empirical probabilities are used reference to the cumulative frequency density (CDF) must be avoided. It also must be specified that it refers to deciles 1, respectively 9;

Comment 3: The National Climate Center of China developed the CZI (China-Z Index) in 1995 as an alternative to the SPI. Why wasn't this one used which is more appropriate based on a three-parameter distribution, namely Pearson type III.

I recommend using the predefined fonts in the MDPI template.

Congratulations on your results and I agree to publish after a minor revision.

Best Wishes,

The reviewer

Reviewer 3 Report

Article title: Analysis of the Composite Risk Grade for Multi Extreme Cli-mate Events in China in Recent 60 Years

Review report:

The results show that the high value areas of MXCI were mainly located in Southeast China and Southwest China. The MXCI presented a decreasing trend in most of the western and southeastern regions of China, while an increasing trend from Southwest China to the south of Northeast China, due to the frequent occurrence of extreme high temperature, extreme drought and other extreme events in these regions.

This work is publishable in the journal after addressing following issues. 

It is necessary to write a problem identification at the start of abstract. 

Based on the findings, I recommend to write the policy implications in the abstract. 

The last sentence of the the first paragraph of introduction should have to modify with addition of given statements and studies as (1-4) “The combustion of fossil fuels has led to climate change, giving rise to a cascade of extreme weather events. As a result, the production system is now grappling with profound and far-reaching consequences (1-4). Therefore, to proactively cope with frequent extreme weather and climate events and their resultant adverse socioeconomic impacts, the IPCC released a special report ‘Managing the Risks of Extreme Events and Disasters to Advance Climate Change Adaptation (SREX)’ in 2012. 

(1)Extreme weather events risk to crop-production and the adaptation of innovative management strategies to mitigate the risk: A retrospective survey of rural Punjab, Pakistan

(2)Understanding farmers’ intention and willingness to install renewable energy technology: A solution to reduce the environmental emissions of agriculture.

(3)Sensitivity analysis of greenhouse gas emissions at farm level: case study of grain and cash crops

(4)Analysis of Energy Input–Output of Farms and Assessment of Greenhouse Gas Emissions: A Case Study of Cotton Growers

I suggest to add main research questions and methods were used to address those research questions in the introduction. 

To avoid length of article, I suggest removing the structure of article from the introduction. 

The given legends of all figures are not very clear, therefore, I suggest to increase font size of the text. 

I recommend to add limitations of the study and recommends for future studies at the end of conclusion section.

Reviewer 4 Report

As a whole the manuscript is well written and structured. The objectives are clear  and well defined. Results are consistent with the proposed methodology. Only a few clarifications on specific points would help the reader to understand the text more quickly: a) In different parts of the manuscript the following procedure to increase the size of the samples is described: "In order to increase the number of samples, the sequences of 5 days before and after are used to form the sequence of the day, and the new sequence is sorted to determine the thresholds". This procedure is not immediately clear to me and furthermore its adoption should be justified, demonstrating that it does not alter the statistical structure of the data; b) in paragraph 2.8 only equation 9 is described but not 8. Furthermore H should be replaced with R. It is not clear the coerence between Eq(8) and Eq (9). It seems that they make reference to different definitions of MXCI. The definitions of R and MXCI should be better clarified.

Reviewer 5 Report

The authors used station observations in China to prepare and analyze the extreme climate indices for multiple hazards, including high and low temperature, drought, precipitation and typhoon impacts. Then a composite risk grade index is formed to examine the overall climate risk for the region. The study is certainly important for assessing the impacts from multiple hazards and of reference to climate change adaption and mitigation. There are improvements necessary in the manuscript on analyzing the individual hazards' trends as well as clarification on how the composite risk index is formed. After such improvements, the manuscript may be considered for publication. The comments are as follow.

1. In equation 1, explain the parameter M more. I think it is at your selected threshold for CF, the number of sample before that. Also, if this empirical formula is similar to fitting a gamma distribution, you have to justify that it works well for all the distributions of the hazards under analysis.

2. Equation 8 is the composite risk grade index you used, which is a simple sum of those from inidividual hazards. However, you have extensively reviewed in the introduction about the necessity of adding appropriate weights to the hazards according to their relative losses. Why such weights have not been applied in this study?

3. In section 3.2 the trend of the extreme drought index is analyzed. It was mentioned that the trend is not the same as that in previous studies because most of them analyzed up to 2010 and this has been extended to 2021 here. You may justify this by showing the trend during 2010-2021 only and before (e.g., in an appendix), which would help to compare with previous studies.

4. At the end of section 3.3 you discussed the impact of increased water vapor to extreme precipitation. While this is true, it may not be the main reason because there are many other factors contributing to extreme precipitatio, such as atmospheric circulation leading to more severe convective storms, etc. Please revise the discussion.

5. In section 3.4, compare your results on the trends of extreme typhoon precipitation and winds with previous studies. For typhoon landfall, example is Shan and Yu (2021, doi:10.1175/JCLI-D-21-
0031.s1), for typhoon precipitation Liu and Wang (2020, doi:10.1175/JCLI-D-19-0693.1), for extreme precipitation Su et al. (2022, doi:10.1088/2515-7620/ac972a) among others.

6. Also, the methods for separating typhoon and non-typhoon rain and winds in this study have been well recognized. However, there are many typhoon remote rainfall events, such as the 2021 Henan Zhengzhou rainstorm event discussed by the authors. These events are difficult to identify and have not all been identified by the methods here. Discussion on such limitation should be added.

7. In the discussion of the composite risk index, if it is eventually decided to use the unweighted index, it should be emphasized that the MXCI here is just a simple sum of the individual hazards without considering their relative losses. In addition, it should be cautious to use the term compound hazard because usually this refers to multiple hazards that have physical processes leading from one hazard to another. While the hazards analyzed in this study must include some compound events (e.g., high temperature to drought, typhoon to extreme precipitation), their compound processes have not been explicitly identified. This should be emphasized in the final dicussion as well.

N/A

Round 2

Reviewer 5 Report

The authors have responded to my comments sufficiently and have revised the manuscript according. Recommendation to publish it can be made.